# Effect of π-Conjugated Spacer in N-Alkylphenoxazine-Based Sensitizers Containing Double Anchors for Dye-Sensitized Solar Cells

**DOI:** 10.3390/polym13081304

**Published:** 2021-04-16

**Authors:** Yung-Sheng Yen, Velu Indumathi

**Affiliations:** Center for Nanotechnology, Department of Chemistry, Chung Yuan Christian University, Zhongli 32023, Taiwan; indumathichemstry@gmail.com

**Keywords:** phenoxazine, dye-sensitized solar cells, sensitizers

## Abstract

A series of novel double-anchoring dyes for phenoxazine-based organic dyes with two 2-cyanoacetic acid acceptors/anchors, and the inclusion of a 2-ethylhexyl chain at the nitrogen atom of the phenoxazine that is connected with furan, thiophene, and 3-hexylthiophene as a linker, are used as sensitizers for dye-sensitized solar cells. The double-anchoring dye exhibits strong electronic coupling with TiO_2_, provided that there is an efficient charge injection rate. The result showed that the power conversion efficiency of **DP-2** with thiophene linker-based cell reached 3.80% higher than that of **DP-1** with furan linker (η = 1.53%) under standard illumination. The photovoltaic properties are further tuned by co-adsorption strategy, which improved power conversion efficiencies slightly. Further molecular theoretical computation and electrochemical impedance spectroscopy analysis of the dyes provide further insight into the molecular geometry and the impact of the different π-conjugated spacers on the photophysical and photovoltaic performance.

## 1. Introduction

Dye-sensitized solar cells (DSSCs) have cultivated a huge interest owing to its ease of fabrication and low cost since the report by Grätzel et al. in 1991 [1]. They imitate natural photosynthesis by means of a molecular absorber, which transforms light into electrical energy [2]. The high power conversion efficiencies of 13.0%, 11.50%, and 12.5% has been reached for porphyrin- [3], polypyridyl Ru(II) complex- [4], and metal-free organic dye-based [5] DSSCs, correspondingly. Many current reports are seen that suggest metal free organic sensitizers might have power conversion efficiency (PCE) cell efficiencies cheaply with that of Ru-based DSSCs [6]. Although Ru-based complexes maintain the record of validated efficiency of over 11%, the Ru-based dyes are costly and environmental unfriendly. Compared to the Ru dyes, metal-free organic dyes have more than a few favors: (i) they take greater molar extinction coefficients; (ii) simple synthesis; (iii) lower cost and more flexible in molecular design. As such, metal-free dyes are also considered hopeful for applications in DSSCs, and several organic dyes has been used as the sensitizers of DSSCs [7,8,9,10,11]. Besides dye-sensitized solar cells, development of new technologies with renewable energy (e.g., sunlight) for the production of H_2_, CO, CH_4_ solar fuels for fuel cell applications also have attracted attention, since they help effective reductions of energy shortage and environment pollution [12,13].

Organic sensitizers usually construct with a D–π–A structure configuration, comprising of a donor (D), a π-linker (π), and an acceptor (A), which also functions as the anchor. The heteroaromatic rings such as thiophene and furan that have been demonstrated to be useful for red shifting of the absorption wavelength were widely used as π-conjugated spacers in sensitizers. Furan also has lower resonance energy than thiophene, which would favor formation of the quinoid structure during intramolecular charge transfer. For example, Hua et al. reported the benzotriazole-based sensitizers containing furan moiety for DSSCs applications, and the result shows that the dyes containing furan moiety as spacers can improve the device efficiency and long-term stability [14]. Anchoring groups also play an important role whether on electron injection or bonding on the TiO_2_ surface [15]. Traditionally, the most commonly used anchors in DSSCs are carboxylic acid and cyanoacrylic acid groups [16]. Subsequently, in the recent exponent development of DSSCs research, various new anchors have occurred and have been tried, that has expressively increased the choice of variable materials, and facilitates awareness of DSSCs. Previously, we developed a sequence of two-anchored organic sensitizers based on imidazole with moderate solar cell performance [17]. We found that they truly not only efficiently inhibited dark current, but also led to excited-state quenching and hindered electron injection. Development of organic dye with di- or multi anchor groups displayed not only enhanced optical properties, but also an increase of electron extraction paths compared to those of dyes with a single anchor. For example, Kumar et al. designed and synthesized a dihydrophenazine dye with two anchor groups recently [18]. In this case, the result shows that dihydrophenazine dye with double-anchors inhibits the aggregation on TiO_2_ and thus reduces the charge recombination. Zhang et al. reported a series of phenothiazine dye with five anchors, and the result displayed that increasing the number of electron acceptors in the backbone of dyes can improve the light-harvesting ability, electrochemical properties, and suppress dark current [19]. Normally, dyes with two acceptors, D(–π–A)_2_, make it [20,21,22], or might not [23,24], show more efficient dark current suppressing than their D–π–A relative. Consequently, photocurrents can be increased.

Among various electron donors, besides the use of triphenylamine as a donor for sensitizers, phenothiazine and phenoxazine electron donor units featuring unique electronic and optical properties have been introduced into a push–pull organic dye which exhibits promising efficiency. Recently, Liu et al. reported a series of dyes based on thieno [2,3-*f*]benzofuran (BDF), by the incorporation of several donors from triphenylamine, carbazole, and phenothiazine [25]. Among these dyes, employing phenothiazine as a donor showed the highest open-circuit voltage (*V*_OC_), short-circuit photocurrent density (*J*_SC_), and PCE. Phenothiazine has the strongest electron-donating property, which improves the light-harvesting ability of the dye and effectively suppresses the dark current. We similarly found the substituents at the nitrogen atom of phenothiazine-cored dyes of the (D(–π–A)_2_) (D = phenoxazine) kind have a profound effect on the cell performance [6,21,26], such as light-harvesting, dye aggregation, and dark current, and the maximum cell efficiency was compared with that of the N719-based standard cell. Hence, there are a growing number of reports on dyes with two-anchoring groups in fresh years [27,28,29,30,31,32,33]. These D(–π–A)_2_ dyes show an interesting phenomenon that not only displays better light-harvesting ability but also provides efficient electron extraction compared to the congeners with only one anchor. Phenoxazine shows similar structural and electrochemical properties to phenothiazine, which implies that phenoxazine could also be incorporated as a dye sensitizer. With our continuing interest in metal-free sensitizers for DSSC applications, we decided to incorporate the phenoxazine moiety in the conjugation spacer of the dye.

Herein, we report new phenoxazine-based sensitizers with two anchors and used as sensitizers for dye-sensitized solar cells. In this present work, D(–π–A)_2_ type di-anchoring dyes based on phenoxazine central core with different π-conjugated spacers were synthesized and for dye-sensitized solar cells application. The photophysical properties of the DP compounds and the performance of the DSSCs fabricated from these three dyes are also discussed.

## 2. Materials and Methods

### 2.1. General Method

All chemicals commercially available, starting materials and solvents, were purchased from Aldrich (St. Louis, MO, USA), TCI (Tokyo, Japan) or Acros Co. (Fukuoka, Japan), and used as received. Solvents THF, toluene, and diethyl ether were distilled under a nitrogen atmosphere with sodium and benzophenone. Other chemicals were bought and used without further purification.

The ^1^H NMR and ^13^C spectra were recorded using a Bruker Advance NMR 300 Hz spectrometer with CDCl_3_ and DMSO-d_6_ that were purchased from Cambridge Isotope Laboratories Inc. (Tewksbury, MA, USA). Absorption spectra were recorded on a JASCO V-730 probe UV−vis spectrophotometer. All chromatographic separations were carried out on silica gel (45–75 um mesh). Mass spectra (FAB) were verified on a Micro TOF-II mass spectrometer.

The photoelectrochemical characterizations on the solar cells were carried out using an Oriel Class AAA solar simulator (Oriel 94043 A, Newport Corp., Irvine, CA, USA). Photocurrent-voltage characteristics of the DSSCs were recorded with a potentiostat/galvanostat (CHI650B, CH Instruments, Inc., Bee Cave, TX, USA) at a light intensity of 100 mWcm^−2^ calibrated by an Oriel reference solar cell (Oriel 91150, Newport Corp., Irvine, CA, USA). The monochromatic quantum efficiency was recorded through a monochromator (Oriel 74100, Newport Corp., Irvine, CA, USA) at short circuit condition. The intensity of each wavelength was in the range of 1 to 3 mWcm^−2^. Electrochemical impedance spectra (EIS) were recorded for DSSCs in the dark at −0.65 V potential at room temperature, whose frequency travelled ranged from 10 mHz to 100 kHz.

### 2.2. Fabrication of DSSCs

The photoanode used was the TiO_2_ thin film (12 μm of 20 nm particles as the absorbing layer and 6 μm of 100 nm particles as the scattering layer) coated on FTO glass substrate with a dimension of 0.5 × 0.5 cm^2^, and the film thickness measured by a profilometer (Dektak3, Veeco/Sloan Instruments Inc., Santa Barbara, CA, USA). A platinized FTO produced by thermopyrolysis of H_2_PtCl_6_ was used as a counter electrode. The TiO_2_ thin film was dipped into the THF solution containing 3 × 10^−4^ M dye sensitizers for at least 12 h. For the co-adsorbed solar cell, chenodeoxycholic acid (CDCA) was added into the dye solutions at a concentration of 10 mM. After rinsing with THF, the photoanode, adhered with a polyimide tape of 30 μm in thickness and with a square aperture of 0.36 cm^2^, was placed on top of the counter electrode and tightly clipped them together to form a cell. A 0.6 × 0.6 cm^2^ of cardboard mask was clipped onto the device to constrain the illumination area. An electrolyte was then injected into the space and then the cell was sealed with the Torr Seal cement (Varian, MA, USA). The electrolyte was composed of 0.5 M lithium iodide (LiI), 0.05 M iodine (I_2_), and 0.5 M 4-tert-butylpyridine that was dissolved in acetonitrile.

### 2.3. Quantum Chemistry Computation

Q-Chem 4.0 software was used for the computations. B_3_LYP/6-31G* basis set was used for geometry optimization of the molecules. For each molecule, a number of possible conformations were examined and the one with the lowest energy was used. The same function was also applied for the calculation of excited states using time-dependent density functional theory (TD–DFT). There exist a number of previous works that employed TD–DFT to characterize excited states with charge-transfer character. In some cases, underestimation of the excitation energies was seen. Therefore, in the present work, we use TD–DFT to visualize the extent of transition moments as well as their charge-transfer characters, and avoid drawing conclusions from the excitation energy.

### 2.4. Synthesis

All the new dyes were prepared via Knoevenagel condensation reaction by reacted corresponding aldehyde derivatives and cyanoacetic acid in the presence of the catalytic amount of ammonium acetate.

*(2E,2′E)-3,3′-(5,5′-(10-(2-ethylhexyl)-10H-phenoxazine-3,7-diyl)bis(furan-5,2-diyl))bis(2-cyanoacrylic acid)* (**DP-1**). Spectroscopic data for **DP-1**: ^1^H NMR (400 MHz, THF-d_8_): δ 7.93 (s, 2H), 7.47 (d, *J* = 8.0 Hz, 2H), 7.35 (d, *J* = 3.2 Hz, 2H), 7.22 (s, 2H), 6.97 (d, *J* = 4.0 Hz, 2H), 6.83 (d, *J* = 8.0 Hz, 2H), 3.64 (d, *J* = 7.2 Hz, 2H), 1.96–1.89 (m, 1H), 1.46–1.29 (m, 8H), 0.962 (t, *J* = 7.2 Hz, 3H), 0.980 (t, *J* = 3.2 Hz, 3H). ^13^C NMR (400 MHz, THF-d_8_): δ 164.59, 160.18, 148.89, 146.15, 138.03, 135.62, 126.01, 123.60, 122.60, 116.86, 114.16, 113.00, 109.00, 98.04, 71.49, 48.36, 37.84, 31.64, 30.80, 29.83, 27.81, 14.51, 28 11.42. MS-HR-MALDI: [M]^+^ calculated for C_36_H_31_N_3_O_7_: 617.65, found: 617.216

*(2E,2′E)-3,3′-(5,5′-(10-(2-ethylhexyl)-10H-phenoxazine-3,7-diyl)bis(thiophene-5,2-diyl))bis(2-cyanoacrylic acid)* (**DP-2**). Spectroscopic data for **DP-2**: ^1^H NMR (400 MHz, THF-d_8_): δ 8.31 (s, 2H), 7.81 (d, *J* = 4.0 Hz, 2H), 7.44 (d, *J* = 4.0 Hz, 2H), 7.27 (dd, *J* = 8.4, 2.0 Hz, 2H), 7.08 (d, *J* = 2.4 Hz, 2H), 6.78 (d, *J* = 8.8 Hz, 2H), 3.63 (d, *J* = 7.6 Hz, 2H), 1.95–1.89 (m, 1H), 1.56–1.29 (m, 8H), 0.97 (t, *J* = 7.6 Hz, 3H), 0.90 (t, *J* = 7.2 Hz, 3H). ^13^C NMR (500 MHz, DMSO-d_6_): δ 163.71, 152.02, 146.34, 144.22, 141.52, 133.88, 133.45, 125.61, 124.07, 122.56, 116.69, 114.56, 113.62, 112.44, 97.59, 46.71, 36.03, 29.84, 28.08, 23.37, 22.54, 13.83, 10.77. MS-HR-MALDI: [M]^+^ calculated for C_36_H_31_N_3_O_5_S_2_: 649.16, found: 649.17

*(2E,2′E)-3,3′-(5,5′-(10-(2-ethylhexyl)-10H-phenoxazine-3,7-diyl)bis(3-hexylthiophene-5,2-diyl))bis(2-cyanoacrylic acid)* (**DP-3**). Spectroscopic data for **DP-3**: ^1^H NMR (400 MHz, THF-d_8_): δ 8.38 (s, 2H), 7.35 (s, 2H), 7.25 (d, *J* = 7.6 Hz, 2H), 7.06 (d, *J* = 1.6 Hz, 2H), 6.75 (d, *J* = 7.6 Hz, 2H), 3.62 (d, *J* = 6.8 Hz, 2H), 2.83 (t, *J* = 7.6 Hz, 4H), 1.45–1.33 (m, 25H), 0.97 (t, *J* = 7.2 Hz, 3H), 0.90 (t, *J* = 6.4 Hz, 9H). ^13^C NMR (400 MHz, THF-d_8_): δ 164.79, 156.54, 152.21, 146.18, 143.84, 135.47, 129.89, 127.52, 125.93, 123.36, 117.04, 114.07, 113.79, 97.85, 48.36, 37.99, 32.69, 32.27, 31.82,30.03, 29.96, 29.85, 24.13, 23.59, 14.53, 14.49, 11.44. MS-HR-MALDI: [M]^+^ calculated for C_48_H_55_N_3_O_5_S_2_: 817.36, found: 817.40

## 3. Result and Discussion

### 3.1. Synthesis

The structures of new di-anchor-based **DP-1** to **DP-3** organic dyes are shown in Figure 1, and the main synthetic ways are described in Scheme 1. Commercially variable 10*H*-phenoxazine was firstly used for preparation of the key starting compound 10-(2-ethylhexyl)phenoxazine by the similar alkylation procedure as described in literature [34]. 3,7-dibromo-10-(2-ethylhexyl)-10*H*-phenoxazine was obtained by bromination reaction of the 10-(2-ethylhexyl)-10*H*-phenoxazine according to defined procedure [35]. Suzuki coupling of **1** with aldehyde-containing boronic acid reagents provided the corresponding aldehyde precursors **2a** and **2b**. Stille coupling of **1** with 3-hexylthiophene stannyl reagent afforded **3**. Subsequent Vilsmeier–Haack reactions of **3** yielded compound **4**. Finally, Knoevenagel condensation of **2a**, **2b**, and **4** with cyanoacetic acid afforded the desired dyes **DP-1**–**DP-3**.

### 3.2. Optical Properties

The photophysical properties of the three new dyes **DP-1**, **DP-2**, and **DP-3** that were investigated by the UV–vis absorption spectra of dyes in THF are shown in Figure 2a and the consistent data are summarized in Table 1. As shown, **DP-1**, **DP-2**, and **DP-3** exhibited similar absorption bands, having two distinct bands at around 350 nm and 520 nm in THF, respectively. The former was assigned to the localized aromatic π–π* transitions, and the latter was attributed to an intramolecular charge-transfer (ICT) transition from the phenoxazine donor to the anchoring moiety. The ICT absorption maximum peaks of **DP-1**, **DP-2**, and **DP-3** were found at 524, 526, and 528 nm, respectively, while the emission bands were displayed at 636 nm, 653 nm, and 647 nm. The Stokes shifts between the absorption and the emission bands were also supported for transfer characteristics in these dyes. Figure 2b shows the photoluminescence spectrum and the consistent data are summarized in Table 1. The maximum absorption peaks of **DP-2** and **DP-3** are slightly red-shifted compared to that of **DP-1** with furan linker. It indicated that the delocalization of electrons over whole molecules with different π-spacers decreased in the order of n-hexylthiophene > thiophene > furan. In addition, the molar extinction coefficients (ε) of these ICT bands significantly increased in the order of **DP-1** (1.5 × 10^4^ M^−1^cm^−1^) > **DP-2** (4.2 × 10^4^ M^−1^cm^−1^) > **DP-3** (4.8 × 10^4^ M^−1^cm^−1^). The organic dyes have higher molar extinction coefficients that were beneficial for device fabrication afforded to use thinner TiO_2_ film. Amongst these dyes, **DP-3** exhibits the broadest and the most intense absorption spectra.

The absorption spectra of the three dyes (**DP-1**, **DP-2**, and **DP-3**) adsorbed onto TiO_2_ films with and without CDCA are shown in Figure 3. Generally, the absorption maxima of organic dyes on TiO_2_ films would change due to the effect of the deprotonation in the adsorption process and the aggregation state of dyes on TiO_2_ films. When **DP-1**, **DP-2**, and **DP-3** adsorbed on the TiO_2_ surface, the absorption spectra of the three dyes were broadened and absorption bands at the long-wavelength side are blue-shift compared to that of the solution spectra, which could be ascribed to the H-type aggregation or the deprotonation of the carboxylic acid upon being adsorbed on TiO_2_ [36,37]. In addition, it was distinguished that the absorption spectra of the three dyes anchored onto the TiO_2_ film exposed a slightly broad outline compared to those in solution, which was helpful for light-harvesting. CDCA was added to check the dye aggregation with DP dyes on TiO_2_ film, as displayed in Figure 3.

After the addition of CDCA, there was no obvious change for λ_max_ of the three dyes, but the absorption intensity showed a decrease for **DP-1** and **DP-2**, which may be due to increased surface coverage of TiO_2_ with CDCA. In contrast, **DP-3**, with hexyl-thiophene linker, displayed a slightly decreased absorption intensity when 10 mM CDCA was added, indicating that the hexyl substituent of the thiophene entity helps with suppression of dye aggregation.

### 3.3. Electrochemical Properties

The energetic arrangement of the highest occupied molecular orbital (HOMO) and lowest unoccupied molecular orbital (LUMO) energy levels is fundamental for an efficient operation of the organic sensitizer in DSSCs. The electrochemical properties of **DP-1** to **DP-3** were analyzed by cyclic voltammetry in THF solution. The representative cyclic voltammograms of the dyes are shown in Figure 4 and the relevant electrochemical data are presented in Table 1. All redox potentials are referenced to ferrocene utilized as an internal standard for calibrating the potential and calculating the HOMO levels. The excited state potential (E_0−0*_) of the sensitizer was estimated from the first oxidation potential (E_ox_) at the ground state and the zero−zero excitation energy (E_0−0_) estimated from the absorption onset. The assumed E_0−0*_ values (−1.10 to −1.09 V vs. NHE, see Table 1) are more negative than the conduction band edge energy level of the TiO_2_ electrode (−0.5 V vs. NHE) [38], and the first oxidation potentials of the dyes **DP-1**, **DP-2**, **DP-3** were measured to be 1.02, 1.01, and 1.00 V vs. NHE are more positive than the I^−^/I_3_^−^ redox couple (~0.4 V vs. NHE) [39]. These results ensure favorable electron injection upon photoexcitation and regeneration of the dye after electron injection.

### 3.4. Photovoltaic Properties

The DSSCs of DP dyes were fabricated using these three dyes as photosensitizers and measured under AM 1.5 G irradiation (100 mW cm^−2^). The current density−voltage (*J*−*V*) curves under illumination and in the dark are shown in Figure 5. The photovoltaic performance statistics under a solar condition (AM 1.5) illumination are collected in Table 2. All the devices exhibited power conversion efficiencies ranging from 1.53 to 3.96%. Under the same condition, the power conversion efficiency of reference dye N719-based cell showed 7.38%. DSSCs based on **DP-1** and **DP-3** dyes exhibited overall power conversion efficiencies of 1.53% and 2.92%, with *J*_SC_ of 3.31 mA cm^−2^ and *J*_SC_ = 5.62 mA cm^−2^, *V*_OC_ of 0.64 V and *V*_OC_ of 0.68 V, and FF of 0.72 and FF of 0.76, respectively. In comparison, the device based on **DP-2** gave a short-circuit photocurrent density (*J*_SC_) of 8.14 mA cm^−2^, an open-circuit voltage (*V*_OC_) of 0.68 V, and a fill factor (FF) of 0.69, consistent with an overall conversion efficiency (η) of 3.80%. The *V*_OC_ value of **DP-2** and **DP-3** was 40 mV higher than that of **DP-1**, which could be attributed to the smaller dark current for **DP-2** and **DP-3**. In other words, **DP-1** with furan moiety had the lowest *J*sc value and poor photovoltaic performance. The device efficiencies are in the order of **DP-2** > **DP-3** > **DP-1** and the performances of the devices based on **DP-2** are ~51% of the standard cell based on ruthenium dye **N719**. The adsorbed dye densities of the sensitizers on TiO_2_ were measured to be 3.70 × 10^−7^, 4.2 × 10^−7^, and 2.1 × 10^−7^ mol/cm^2^ for **DP-1**, **DP-2**, and **DP-3**, respectively. The low dye density on TiO_2_ of **DP-3** might be attributed to more steric congestion of n-hexylthiophene linker owing to the presence of an extra n-hexyl chain. The higher cell efficiency of **DP-2** than **DP-3** and **DP-1** is ascribed to the better light-harvesting of **DP-2** and its slightly higher dye density amount on TiO_2_ film compared to the other two dyes. **DP-2** has better light-harvesting efficiency in the film among the three DP dyes; it also has greatly higher incident monochromatic photo-to-current conversion efficiency (IPCE) values than the other two dyes in the range of 400–700 nm. The result indicates that **DP-2** has the highest photocurrent due to higher dye absorption density on TiO_2_ and faster and more effective electron injection efficiency.

The incident monochromatic photo-to-current conversion efficiency (IPCE) plots of the cells are shown in Figure 6, respectively. Well-consistent with adsorption spectra in TiO_2_, **DP-2** exhibited broader and higher IPCE efficiencies. The IPCE value for **DP-2** at around 520 nm (60%) is higher than that of **DP-1** (39%) or **DP-3** (17%). IPCE is related to the light-harvesting efficiency of the photoelectrode and electron injections yield and charge collection efficiency. The higher IPCE values of the DSSC devices based on **DP-2** or **DP-3** with thiophene or hexylthiophene linkers have higher electron transfer yield than the dye with furan linker. This indicated that the introduction of thiophene heteroaromatic ring into phenoxazine-based sensitizer structure has a positive effect.

Suppression of dye aggregation was also supported by the blue shift of the absorption spectra of the DP dyes on the TiO_2_ film when CDCA was added. The cell performance data with CDCA co-adsorbent are summarized in Table 2. The three dyes have slight improvements in the cell performance upon addition of CDCA: PCE = 1.80%, *J*_SC_ = 3.64 mAcm^−2^, *V*_OC_ = 0.65 V, FF = 0.76 for **DP-1**; PCE = 3.96%, *J*_SC_ = 8.52 mAcm^−2^, *V*_OC_ = 0.69 V, FF = 0.67 for **DP-2**; PCE = 3.10%, *J*_SC_ = 6.23 mAcm^−2^, *V*_OC_ = 0.68 V, FF = 0.74 for **DP-3**. For **DP-1** to **DP- 3**, the cell performance improved only marginally upon addition of CDCA 10 mM, *V*_OC_ stayed almost the same, whereas *J*_SC_ continued to increase. Hence, anti-aggregation of the dyes was alleviated by CDCA adsorption. By adding 10 mM CDCA, devices with **DP-2** showed the better *J*_SC_, *V*_OC_, and conversion efficiency of 3.96% (Table 2).

### 3.5. Electrochemical Impedance Spectroscopy Analysis

The electrochemical impedance spectroscopy (EIS) is a very useful technique to understand the electron injection and recombination processes in DSSCs [40]. Electrochemical impedance spectroscopy (EIS) was used to further evaluate the important interfacial charge transfer processes in a DSSC. Generally, there are three semicircles showed in the EIS spectrum, which correspond to electron recombination resistances (Rrec) at the interfaces of photoanode/dye/electrolyte, charge-transfer resistance at the photoanode/dye/electrolyte interface (RCT), and Warburg diffusion process of electrolyte (Zw), as typically reported for other DSSC devices [41,42,43]. The electrochemical impedance spectra (EIS) of DSSCs were obtained under a forward bias of −0.70 V in the dark to elucidate correlation of *V*_OC_ with those dyes, and the Nyquist plots for DSSCs based on **DP-1** to **DP-3** are shown in Figure 7. The large semicircle in the Nyquist plots is attributed to the charge recombination resistance between the TiO_2_ and the electrolyte (R_rec_), where the larger R_rec_ value suggests the smaller dark current. The radius of the biggest semicircle increases in the order of **DP-1** < **DP-3** < **DP-2**. The cell of **DP-2** and **DP-3** exhibits a much larger resistance value than that of **DP-1**, which is consistent with its smaller dark current and larger *V*_OC_ measured.

### 3.6. Computational Calculation

In order to gain insight into the relationship between the geometrical and electronic properties of the **DP** dyes, the dyes **DP-1** to **DP-3** were further investigated through theoretical calculations. The results for theoretical computation are included in Table 3. Figure 8 shows the ground-state geometries of the dyes with the dihedral angles between the two neighboring conjugated segments indicated. In the optimized structure, **DP-1** has a nearly planar structure where the torsion angle between phenoxazine and the furan entity is almost 0°. The planarity structure of **DP-1** can increase the stacking of the dye molecules, inducing more dye aggregation. In comparison, the dihedral angle between the phenoxazine and the thiophene entity is larger than 20° for **DP-2** and **DP-3**. The smaller planarity of **DP-2** and **DP-3** leads to better charge separation between the phenoxazine unit and accepter unit. The electron distributions of the HOMOs and LUMOs for **DP-1**, **DP-2**, and **DP-3** are illustrated in Figure 9. The HOMOs of **DP-1**, **DP-2**, and **DP-3** are delocalized on the entire molecule including acceptor, whereas the LUMO and LUMO+1 of these molecules are mainly distributed from the π-spacer acceptor to the acceptor. In Table 3, the S_0_ → S_1_ transition is nearly a HOMO → LUMO transition. Therefore, the lowest energy absorption has charge transfer character for these dyes. The more intense electronic absorptions in **DP-2** and **DP-3** than **DP-1** are supported by its larger computed oscillation strength (f). The Mulliken charges variation for the S_1_ and S_2_ states were calculated from the TD-DFT results. Differences in the Mulliken charges in the excited and the ground states were calculated and gathered into several segments, heteroaromatic ring (F, T, T_1_), phenoxazine (Poz), heteroaromatic ring (F’, T’, T_1′_), and 2-cyanoacrylic acid (Ac), 2-cyanoacrylic acid (Ac’) to estimate the extent of charge separation upon excitation. Figure 10 displays the changes in Mulliken charges of the dyes for the S_0_ → S_1_ and S_0_ → S_2_ transitions. In **DP-1** to **DP-3**, the positive charges exist at phenoxazine in the dyes for S_0_→S_1_ and S_0_→S_2_ transitions. On the other hand, **DP-1** to **DP-3** have prominent negative charges at both acceptors for both S_0_→S_1_ and S_0_→S_2_ transitions, indicating that both acceptors can function as the electron injection channels.

## 4. Conclusions

In summary, we reported the new phenoxazine-based organic dyes containing a 2-ethylhexyl substituent at the nitrogen atom of the phenoxazine and two 2-cyanoacrylic acids as the acceptors in addition to anchors. The influences of the various π bridges on the photophysical, electrochemical, and photovoltaic properties of these sensitizers were investigated. The results of calculation and experiments clearly demonstrate that photophysical properties can be regulated by introducing different π-conjugated bridges. Then, the performance of the DSSCs based on these dyes were tested and analyzed, and, among which, the **DP-2** cell shows the best PCE of 3.80% without CDCA among all. DSSCs using these three **DP** dyes as the sensitizers showed efficiencies ranging from 1.53 to 3.80% without CDCA under simulated AM1.5G irradiation. Upon addition of CDCA as a co-adsorbent, the cell efficiency has been further improved to 3.96% for DSSCs based on **DP-2**, which is about 54% of the N719-based standard cell. Our future work will focus on optimization of molecular structure to fine-tune the energy levels of the dye toward higher *V*_OC_, *J*_SC_, and panchromatic DSSCs.

## Data Availability

The data presented in this study are available on request from the corresponding author.

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
