# Peer review of "Effect of π-Conjugated Spacer in N-Alkylphenoxazine-Based Sensitizers Containing Double Anchors for Dye-Sensitized Solar Cells"

_polymers, 2021, doi:10.3390/polym13081304_

Round 1
Reviewer 1 Report
Dye-sensitized solar cells (DSSCs) has fascinated huge interest owing to it is easy to fabricate and low cost since the report by Grätzel et al. in 1991. They imitate the natural photosynthesis by means of a molecular absorber, which was transforms light into electrical energy. In this paper, A series of novel double-anchoring dyes for phenoxazine-based organic dyes with two 2-cyanoacetic acid acceptors/anchors, and the inclusion of a 2-ethylhexyl chain at the nitrogen atom of the phenoxazine that were connected with furan, thiophene and 3-hexylthiophene as linker used as sensitizers for dye-sensitized solar cells. The double-anchoring dye exhibits strong electronic coupling with TiO2, provided that an efficient charge injection rate. The results and theme of this paper is quite interesting. The layout is clear and easy to understand. Generally, this manuscript makes fair impression and my recommendation is that it merits publication in this Journal, after the following major revision:
- The whole introduction has presented a number of related works as literature review. However, the authors simply repeat what those works have done, instead of introducing the brief feature and the coherence of them. Why their work is important comparing to previous reports? I think this is essential to keep the interest of the reader.
- In Figs. 2 and 3, the authors should give the explanations for the difference of data collected from different sources.
- The result showed that the power conversion efficiency of DP-2 with thiophene linker based cell reached 3.80% higher than that of DP-1 with furan linker (η = 1.53%) under standard illumination. The authors should give some explanation on above conclusions and data.
- Materials and Methods part, Although the results look “making sense”, the current form reads like a simple lab report. The authors should dig deeper in the results by presenting some in-depth discussion.
- Energy shortage and environment pollution have seriously threatened people’s survival. Thus, the development of fuel cell has caught human attention. Besides dye-sensitized solar cells, proton exchange membrane fuel cells have attracted attention from energy devices such as portable, mobile and stationary devices, since it helps effective reductions of energy shortage and environment pollution, see [International Journal of Hydrogen Energy, 2018, 43(37):17880-17888; International Journal of Heat and Mass Transfer, 2019, 137:365-371]. Authors should introduce some related knowledge to readers. I think this is essential to keep the interest of the reader.
- Please, expand the conclusions in relation to the specific goals and the future work.
Author Response
Reviewer 1
We thank the reviewer for very valuable remarks to our manuscript.
Specific comments/suggestions:
1. The whole introduction has presented a number of related works as literature review. However, the authors simply repeat what those works have done, instead of introducing the brief feature and the coherence of them. Why their work is important comparing to previous reports? I think this is essential to keep the interest of the reader.
Our response:
We have added more sentences to introducing the related works and explain why their work is important in “Introduction” Please see page 1, line 35-39, line 42-44, page 2, line 45-50, line 57-66.
- In Figs. 2 and 3, the authors should give the explanations for the difference of data collected from different sources.
Our response:
We have already explain the difference between the Figs.2 and Figs. 3 on page 6, lines 229-233.
- The result showed that the power conversion efficiency of DP-2 with thiophene linker based cell reached 3.80% higher than that of DP-1 with furan linker (η = 1.53%) under standard illumination. The authors should give some explanation on above conclusions and data.
Our response:
We have gave an explanation on the page 7, lines 284-287.
- Materials and Methods part, Although the results look “making sense”, the current form reads like a simple lab report. The authors should dig deeper in the results by presenting some in-depth discussion.
Our response:
Materials and Methods part is like the experimental section, providing general information about materials used in the synthesized procedure and general method to characterized and how to obtained the data. We add some sentences and revised errors in the “Fabrication of DSSCs” of Materials and Methods, please see page 3, line 122-123; line 126-127. We believe that the current form is enough.
- Energy shortage and environment pollution have seriously threatened people’s survival. Thus, the development of fuel cell has caught human attention. Besides dye-sensitized solar cells, proton exchange membrane fuel cells have attracted attention from energy devices such as portable, mobile and stationary devices, since it helps effective reductions of energy shortage and environment pollution, see [International Journal of Hydrogen Energy, 2018, 43(37):17880-17888; International Journal of Heat and Mass Transfer, 2019, 137:365-371]. Authors should introduce some related knowledge to readers. I think this is essential to keep the interest of the reader.
Our response:
We have added some sentences in ” Introduction” to introduce related knowledge and cited mention references. please see page 1, line 35-39.
- Please, expand the conclusions in relation to the specific goals and the future work.
Our response:
We have added one sentence related to the specific goals and the future work in the conclusion, please see page 13, line 392-394.

Reviewer 2 Report
Authors present a study on the synthesis and characterization of double-anchoring dyes for phenoxazine-based organic dyes with two 2-cyanoacetic acid acceptors/anchors, and the inclusion of a 2-ethylhexyl chain at the nitrogen atom of the phenoxazine connected with furan, thiophene and 3-hexylthiophene as linker used as sensitizers for dye-sensitized solar cells.
The topic is of scientific soundness and current interest in the field of materials science, particularly for solar cell applications. The manuscript is well-written and exhibits an interesting joint theoretical and experimental approach.
In my opinion, it worth publication in "Polymers" after considering minor revisions shown below:
1) The introduction section omits relevant literature similar to the one presented here. To mention a few:
10.1021/acsomega.0c01255
10.1016/j.solener.2020.11.008
10.1002/jccs.201900410
10.1002/asia.201100967
10.1038/s41598-017-05408-8
Authors must refer to this literature and explain their contribution to the state of the art further.
2) Figure 2a & b must be corrected.
Quote (Page 4, line 167) "...The ICT absorption maximum peaks of DP-1, DP-2 and DP-3 were found at 524, 526, and 528 nm, respectively, while the emission bands were displayed at 636 nm, 653 nm and 647 nm..."
However, the plots shown in Figure 2 a & b are exactly the same, both seem to belong the absorption spectra, please add the correct plot for emission spectra in Figure 2b.
3) Figure 5 must be improved.
Please, use other symbology to easily discriminate between J-V and dark currents plots.
4) Further discussion is needed in section 3.5.
Quote (Page 8, line 293) "...The Large semicircle in the Nyquist plots is attributed to the charge recombination resistance between the TiO2 and the electrolyte (Rrec), where the larger Rrec value suggests the smaller dark current..."
The impedance semicircle arc is not entirely attributed to the recombination resistance, it could be the major contribution but need to be further discussed. In fact, impedance data shown in Figure 7 is not based on a perfect semicircle arc (real part is significatively lower than the impedance part) but rather likely to be composed of more contributions. Impedance response of DSSC devices are far more complex to analyze, authors can refer to:
10.1007/s10854-019-00929-6
10.1002/ijch.201500007
10.1016/j.electacta.2015.04.149
10.1016/j.materresbull.2018.05.029
Please briefly discuss the validity of their reported resistance values.
5) Revise the following statement:
Quote: (Page 8, line 296) "...The cell of DP-2 and DP-3 exhibits a much larger resistance value than that of DP-1 which is in parallel with its smaller dark current and larger VOC measured..."
Isn't the expected trend the opposite? please revise.
Author Response
Reviewer 2
Comments:
1) The introduction section omits relevant literature similar to the one presented here. To mention a few:
10.1021/acsomega.0c01255
10.1016/j.solener.2020.11.008
10.1002/jccs.201900410
10.1002/asia.201100967
10.1038/s41598-017-05408-8
Authors must refer to this literature and explain their contribution to the state of the art further.
Our response:
We have cited the mention references and added more sentences to explain their contribution in introduction section. Please see page 1, line 35-39, line 42-44, page 2, line 45-50, line 57-66.
2) Figure 2a & b must be corrected.
Our response:
We have replaced the corrected figure 2b.
3) Figure 5 must be improved.
Please, use other symbology to easily discriminate between J-V and dark currents plots.
Our response:
We have used other symbology in Figure 5.
4) Further discussion is needed in section 3.5.
Quote (Page 8, line 293) "...The Large semicircle in the Nyquist plots is attributed to the charge recombination resistance between the TiO2 and the electrolyte (Rrec), where the larger Rrec value suggests the smaller dark current..."
The impedance semicircle arc is not entirely attributed to the recombination resistance, it could be the major contribution but need to be further discussed. In fact, impedance data shown in Figure 7 is not based on a perfect semicircle arc (real part is significatively lower than the impedance part) but rather likely to be composed of more contributions. Impedance response of DSSC devices are far more complex to analyze, authors can refer to:
10.1007/s10854-019-00929-6
10.1002/ijch.201500007
10.1016/j.electacta.2015.04.149
10.1016/j.materresbull.2018.05.029
Please briefly discuss the validity of their reported resistance values.
Our response:
We have cited the mention references and added more sentences to discuss briefly. please see page 9, line 318-319; line 321-325.
5) Revise the following statement:
Quote: (Page 8, line 296) "...The cell of DP-2 and DP-3 exhibits a much larger resistance value than that of DP-1 which is in parallel with its smaller dark current and larger VOC measured..."
Isn't the expected trend the opposite? please revise.
Our response:
Page 8, line297 “in parallel” has been changed to “consistent”. Please see P9 line 332.
We hope that the revision is found satisfactory.

Round 2
Reviewer 1 Report
In Ref. 14, “Liang, M.; Liu, Y.; Xiao, B.; Yang, S.; Wang, Z.; Han, H. Int. J. Hydrogen Energy 2018, 43, 17880−17888” should be corrected as “Liang, M.; Liu, Y.; Xiao, B.; Yang, S.; Wang, Z.; Han, H. An analytical model for the transverse permeability of gas diffusion layer with electrical double layer effects in proton exchange membrane fuel cells. Int. J. Hydrogen Energy 2018, 43, 17880−17888”